# Microbial Diversity and Community Structure of Wastewater-Driven Microalgal Biofilms

**DOI:** 10.3390/microorganisms11122994

**Published:** 2023-12-16

**Authors:** Olga Blifernez-Klassen, Julia Hassa, Diana L. Reinecke, Tobias Busche, Viktor Klassen, Olaf Kruse

**Affiliations:** 1Algae Biotechnology and Bioenergy, Faculty of Biology, Center for Biotechnology (CeBiTec), Bielefeld University, Universitätsstrasse 27, 33615 Bielefeld, Germany; olga.blifernez@uni-bielefeld.de (O.B.-K.); viktor.klassen@uni-bielefeld.de (V.K.); 2Center for Biotechnology (CeBiTec), Bielefeld University, Universitätsstrasse 27, 33615 Bielefeld, Germanytobias.busche@uni-bielefeld.de (T.B.); 3Institute of Bio- and Geosciences, Plant Sciences, Forschungszentrum Jülich, Wilhelm-Johnen-Strasse, 52428 Juelich, Germany; d.reinecke-levi@fz-juelich.de; 4Medical School East Westphalia-Lippe, Bielefeld University, Universitätsstrasse 27, 33615 Bielefeld, Germany

**Keywords:** environmental microbiome structure, wastewater, taxonomic profiling, biofilm, microbial biodiversity, microalga–bacteria consortia, algal turf scrubber (ATS)

## Abstract

Dwindling water sources increase the need for efficient wastewater treatment. Solar-driven algal turf scrubber (ATS) system may remediate wastewater by supporting the development and growth of periphytic microbiomes that function and interact in a highly dynamic manner through symbiotic interactions. Using ITS and 16S rRNA gene amplicon sequencing, we profiled the microbial communities of four microbial biofilms from ATS systems operated with municipal wastewater (mWW), diluted cattle and pig manure (CattleM and PigM), and biogas plant effluent supernatant (BGE) in comparison to the initial inocula and the respective wastewater substrates. The wastewater-driven biofilms differed significantly in their biodiversity and structure, exhibiting an inocula-independent but substrate-dependent establishment of the microbial communities. The prokaryotic communities were comparable among themselves and with other microbiomes of aquatic environments and were dominated by metabolically flexible prokaryotes such as nitrifiers, polyphosphate-accumulating and algicide-producing microorganisms, and anoxygenic photoautotrophs. Striking differences occurred in eukaryotic communities: While the mWW biofilm was characterized by high biodiversity and many filamentous (benthic) microalgae, the agricultural wastewater-fed biofilms consisted of less diverse communities with few benthic taxa mainly inhabited by unicellular chlorophytes and saprophytes/parasites. This study advances our understanding of the microbiome structure and function within the ATS-based wastewater treatment process.

## 1. Introduction

Environmental water pollution due to anthropogenic activities and the scarcity of water resources in general have underlined the urgent requirement for sustainable wastewater treatment [1]. Because existing conventional treatment systems are often complex and energy- and cost-intensive, algal–bacterial systems have emerged as environmentally friendly, sustainable processes for wastewater treatment, resource recovery, and biomass production [2,3].

Microalgae can grow on nonarable land using fresh and saline water, as well as wastewater and produce large amounts of lipids, proteins, and carbohydrates, which can be processed into valuable feed and coproducts, as well as fertilizers, bioenergy, or biofuels (e.g., biogas, biodiesel, biomethane) [4,5,6,7,8]. Microalgae are unicellular eukaryotic (and in the case of cyanobacteria, prokaryotic) microorganisms that are frequently described as “lower” plants that lack true stems, roots, and leaves [9]; however, many taxa are known to aggregate and form granules, colonies, chains, and filaments [2,10,11]. Within aquatic ecosystems, microalgae are the dominant primary producers and the base of the food web [12] because of their ability to grow photoautotrophically by performing oxygenic photosynthesis using sunlight, CO_2_, and inorganic nutrients. In nature, most algae live in symbiosis with multiple associated microorganisms within their phycosphere [12], and recent observations indicate that algal–bacterial cocultivation strategies offer enormous advantages for biotechnological applications [13,14,15]. Interactions between microalgae and their symbiotic partners are often multifaceted and complex, and their interrelationships within a given phycosphere span from cooperative to competitive, with the nature of the exchange of micro- and macronutrients, various metabolites, and infochemicals defining the relationship [11,12,16,17].

Nowadays, algae–bacteria consortia are considered superior for nutrient removal in wastewater treatment processes compared to conventional systems because of their manifold metabolic capabilities [2,18]. Microalgae utilize CO_2_ for photosynthesis, assimilate nutrients, and release oxygen, which can be used for metabolism by heterotrophic microorganisms for oxidizing organic matter. Inorganic carbon, as well as nitrogen and phosphorus released during bacterial metabolism can be utilized by microalgae [19]. Pollutants in wastewater can be removed by microbial consortia through several processes such as assimilation (uptake of nitrogen and phosphorus), stripping (ammonia removal at high pH), nitrification–denitrification, oxidation of organic carbon to carbon dioxide, and adsorption (heavy metals removal), phosphorus precipitation, and pathogen removal due to pH fluctuations [20]. Along with nutrient removal, the algal–bacterial consortium is also capable of removing micropollutants, pesticides, heavy metals, polycyclic aromatic hydrocarbons, pharmaceuticals, and microplastics [18].

In general, there are two types of algal–bacterial treatment systems: suspended growth and attached growth systems [21]. The suspended growth system is the most commonly used approach for microalgae-mediated wastewater treatment, applying high-rate algal ponds (HRAP) or raceway ponds and photobioreactors (PBRs). Although high nitrogen removal efficiencies (80–100%) are achievable in these systems, they suffer from certain limitations, such as the requirement of larger operation areas and poor biomass settleability, which hinder the achievement of total suspended solids (TSS) disposal standards [22,23]. The attached growth systems may present alternative and cost-effective growth systems for the wastewater treatment process, such as hybrid biofilm photobioreactors, membrane aerated biofilm reactors (MABR), and algal turf scrubbers (ATS) [2,18,21,24]. In particular, ATS technology was developed to promote the natural remediation process of fluvial biofilms [25,26,27]. An ATS system consists of a long, sloping bed that supports biofilm formation of the so-called periphyton [10], which consists of a mixed benthic community that includes microalgae, bacteria, fungi, metazoans, and detritus. During biofilm development, sticky extracellular polymeric substances (EPS) secreted by certain bacteria and microalgae wrap around the cell surface and attach to the carrier, thereby causing the periphytic microorganisms to gather [28,29]. (Waste)water flows down the biofilm, while nutrients and pollutants are removed by the biomass, which is occasionally scraped off for harvesting [8,27]. The periphytic community existing in an ATS can positively influence water quality (e.g., by decomposing nutrients and organic contaminants) due to complex ecological interactions. These interactions are in turn influenced by many external parameters (e.g., sunlight, wastewater composition) and microenvironmental changes due to community growth and metabolism [2,30]. Although the structure and dynamics of the ATS periphyton (microbial communities) are undoubtedly of crucial importance to overall ATS performance, little attention has been paid to them because most ATS-based research has focused on nutrient removal capacities under different wastewater sources.

In this study, we investigated the taxonomic diversity and composition of the prokaryotic and eukaryotic communities for four differently operated ATS systems, the initial inocula, and the respective wastewater substrates using a high-throughput ITS and 16S rRNA gene amplicon sequencing approach. The main research objectives were to characterize the periphytic communities of ATS-based microalgae cultivation systems operating with four different wastewaters: municipal wastewater, diluted cattle and pig manure, and supernatant from a biogas plant. Similarities and differences in the microbiome profiles will provide insights into how wastewater quality modulates community structure and how the resulting composition may affect biomass productivity.

## 2. Materials and Methods

### 2.1. Sample Sources, Sample Collection, and DNA Isolation

For the 16S and ITS rRNA amplicon analyses, microalgae–bacteria-containing biofilms were cultured in in-house-designed ATS systems. Four ATS were operated as described earlier at a farm (51.634 N 8.875 E) and a wastewater treatment plant (WWTP, (51.604 N 8.767 E) with municipal WW (mWW, tertiary wastewater), biogas effluent (BGE), pig manure (PigM), and cattle manure (CattleM), respectively [8]. Initial characterization of the four wastewater samples was performed by a certified external lab according to the German fertilizer regulation (DüMV, Germany). Details of the methods and ranges can be found under DIN EN ISO 11885 2009–09 [31], DIN 38406-E5-2 1983–10 [32], VDLUFA II.1 11.5.1 1995–05, and 3.5.2.7 1995-01 [33]. All ATS were inoculated with preexisting biofilms: Inoculum I from the research center FZ Jülich (50.906 N 6.411 E) and Inoculum II from the local WWTP (51.604 N 8.767 E) in spring 2021 (calendar week 16). The ATS biofilms were harvested in a representative week (calendar week 38), when the biofilms had been established and the overall operating performance of all ATS systems was stable. Harvesting was accomplished by interrupting water flow, draining, and scraping the biofilm. The total ATS biomass productivity was analyzed and calculated as described previously [8,34].

All samples for 16S and ITS rRNA amplicon analyses were collected in three independent biological replicates. Samples of each wastewater, 50 mL of agricultural wastewater (PigM, CattleM, BGE) and 3 L of municipal wastewater per sample, were centrifuged at 16,000× *g* for 30 min, and the respective pellet was frozen in liquid N_2_ and lyophilized. The individual scraped biofilms were frozen in liquid N_2_, lyophilized, and homogenized using a Precellys homogenizer (3× at 6500 rpm for 30 s, Peqlab, Erlangen, Germany) to ensure uniform disruption and mixing of samples. Genomic DNA was extracted as previously described by Zhou et al. [35].

### 2.2. Amplicon Library Preparation and Sequencing

The 16S rRNA gene and ITS libraries were separately constructed from the same DNA samples using the “16S Metagenomic Sequencing Library Preparation” protocol (Illumina Inc., San Diego, CA, USA). For the 16S rRNA gene amplicon libraries, the universal primer pair 341F and 785R [36] was used for amplification of the V3–V4 hypervariable region of the 16S rRNA gene. For the internal transcribed spacer (ITS) amplicon libraries, the universal primer pair ITS1 and ITS2 [37] was used for amplification of the ITS region between the 18S and 5.8S rRNA genes. The constructed libraries were equimolar pooled and subsequently sequenced on the Illumina MiSeq platform using the protocol for 2 × 300 bp paired-end reads. The raw 16S rRNA gene and ITS sequencing data were uploaded to the NCBI Sequence Read Achieve (SRA) database with access number PRJNA1045004.

### 2.3. Bioinformatic Processing

The raw sequences were preprocessed using an in-house pipeline. Briefly, the reads were merged with flash [38], the 16S rRNA gene and ITS primer were removed using cutadapt version 1.8.1 [39], the reads were trimmed by quality using sickle version 1.33 [40], and the quality of the preprocessed reads was analyzed using FastQC [41]. After preprocessing, the QIIME 2 platform (version 2021.8) was used for further processing of the datasets [42]. Firstly, the datasets were denoised using DADA2, version 1.26 [43], and the amplicon sequence variants (ASVs) were aligned using mafft version 6.240 [44]. Afterwards, an unrooted and rooted phylogenetic tree was calculated using fasttree version 2.1 [45]. The taxonomic assignment of the amplicon sequences was performed using the q2-feature-classifier [46] classify-sklearn [47] against the Silva database (release 138, [48]) for the 16S rRNA gene datasets and the Unite (version 8.0, [49]) database for the ITS datasets. The datasets were filtered to eliminate all ASVs with frequencies below five and rarefied to a given depth within each dataset. In contrast, the 16S rRNA gene datasets were subsampled to 50,000 sequences and the ITS datasets to 100,000 sequences. The resulting taxonomic profiling tables were normalized to 100% and visualized as bar charts. The phylum names of prokaryotes were assigned according to the new taxonomy proposed by Oren and Garrity [50]. Statistical analysis for the evaluation of the most abundant taxa at the genus level was performed using the two-tailed Student’s t-test, resulting in *p*-values indicated by coloring (*p* ≤ 0.05 = light green, *p* ≤ 0.01 = dark green). Alpha and beta diversity metrics were calculated using Shannon diversity indices and weighted and unweighted UniFrac distances. The calculated metrics were used in R to visualize tree diagrams and ordination plots.

## 3. Results

### 3.1. Physiochemical Properties of the Wastewater

This study investigated the taxonomic diversity of ATS-based microalgae cultivation systems using four different wastewaters, such as municipal wastewater (mWW), diluted cattle and pig manure, and supernatant from a biogas plant effluent (CattleM, PigM and BGE, respectively). Analysis of the four wastewaters revealed a comparatively low nutrient content in mWW compared with the agricultural wastewater (Table 1). PigM, CattleM and BGE contained up to 28–64 times more nitrogen (N), up to 30–61 times more phosphorus (P), and up to 9–22 times more potassium (K). While ammonium (NH_4_-N) represented the dominant nitrogen form in agricultural wastewater with up to 83% of total N, the mWW was predominated by nitrate (NO_3_-N). However, the N:P molar ratio between the respective wastewaters was relatively comparable with values between 4.7 and 6.0 (Table 1).

The organic content of the wastewater also showed significant differences among the four substrates. Here, the chemical oxygen demand (COD) of mWW had the lowest value with <15 mg L^−1^, followed by PigM, CattleM, and BGE with 1.8, 2.6, and 5.3 g L^−1^, respectively, indicating a more than 100-fold higher content of suspended solids in the agricultural wastewater (Table 1). The applied wastewaters generally differed in their composition and properties, with the most significant differences between mWW and agricultural wastewaters occurring in terms of low or high nutrient content. In general, however, the wastewater used was not limited in nutrients and was therefore very suitable for algae growth [51].

Four ATS systems were inoculated with equal parts of two biofilms containing different microalgal–bacterial communities originating from the research c enter Jülich (FZJ, Inoculum I) and the local environment at the WWTP Lichtenau (Inoculum II). The four ATS were supplemented with either mWW, PigM, CattleM, or BGE wastewater substrates and, after establishment and stabilization of the growth process, were subjected to analysis through 16S and ITS rRNA amplicon sequencing.

### 3.2. Biodiversity of the Different Wastewater-Driven Microbiomes

To profile the microbial communities of the different ATS systems, the sequencing of the samples (in triplicates) obtained from Inoculum I and II, wastewater substrates, and biofilm communities resulted in approximately 4.7 and 2.7 million prokaryotic and eukaryotic sequences, respectively (Appendix A). The rarefaction analysis indicated that the sequencing depth was sufficient to cover the prokaryotic and eukaryotic microbial diversity of all samples (Appendix A). After multiple filtering and processing steps, approximately 77% and 96% of the high-quality data with 3.6 and 2.5 million prokaryotic and eukaryotic sequences remained, respectively. The quality-filtered sequences were clustered into 43,880 and 7795 ASVs as representative sequences, which were then classified into different prokaryotic and eukaryotic taxa. The mWW ATS samples had the highest number with 2135 ± 263 and 792 ± 67 among the prokaryotic and eukaryotic sequences, respectively, whereas Inoculum I with 779 ± 33 ASVs exhibited the lowest value for prokaryotic and PigM ATS with 191 ± 11 among the eukaryotic sequences (Appendix A).

The analyzed samples differed significantly in their biodiversity, as suggested by the Shannon index calculation. Among prokaryotic communities, the highest Shannon index was observed for the mWW substrate with 9.5 ± 0.1, followed by the CattleM substrate and mWW ATS with 8.8 ± 0.1 each, while Inoculum II had the lowest value with 5.8 ± 0.1 (Figure 1, Bacteria). The biodiversity within the eukaryotic community was highest in mWW ATS with 6.4 ± 0.1, followed by Inoculum II and BGE ATS with values up to 5.9. Both Inoculum I and CattleM ATS showed lower diversity indices of up to 4.9, followed by PigM ATS, which displayed the lowest Shannon diversity with 3.6 ± 0.1 among eukaryotes (Figure 1, Eukaryota). Several studies have also reported on the impact of wastewater on the structure of periphyton/microbial communities [52,53,54], where the microbials originating from the respective wastewater are most likely able to colonize the periphytic biofilms, thus likely influencing the overall structure and diversity of the microbial community.

Overall differences in microbial communities between the different ATS systems, wastewater substrates, and initial inocula were statistically assessed (Figure 2). It is apparent that the unweighted UniFrac distance matrix highlights different aspects of the data from weighted UniFrac and separates the subjects into more distinct clusters (Appendix A), likely by unweighting shallow differences.

The grouping of samples based on an unweighted UniFrac distance matrix showed clear clustering of different ATS systems, indicating high similarities within the eukaryotic and prokaryotic community compositions of BGE ATS, CattleM ATS, and PigM ATS, which are distinct from mWW ATS (Figure 2a). Similar observations could be made for the weighted UniFrac distances for the eukaryotic datasets, indicating a differential abundance between the systems operated with agricultural and municipal wastewater. However, for the prokaryotic communities, a less distinct separation could be observed for the same groups, suggesting a higher proportion of common abundant taxa (Figure 2b, Bacteria). The agricultural wastewater substrates (BGE, PigM and CattleM) were clearly separated from the other communities in both weighted and unweighted distance matrices, suggesting significant differences between the groups (Figure 2a,b, Bacteria).

### 3.3. Taxonomic Profiling of the Communities of ATS Biofilms

The taxonomic profiles of the microbiomes of the different ATS systems and their respective wastewater substrates, as well as the initial inocula were determined using 16S rRNA and ITS gene amplicon sequencing. The eukaryotic taxonomic profiles revealed that up to 73%, 34%, and 24% of the detected taxa were assigned to the domains Viridiplantae, Stramenopila, and Fungi, respectively, depending on the corresponding sample (Figure 3). Comparatively high proportions of taxonomically unclassified and unassigned organisms were observed mainly in Inoculum I, followed by mWW ATS with up to 54% and 37%, respectively. Samples of wastewater substrates were not analyzed using ITS gene amplicons because the wastewater was not expected to contain high numbers of eukaryotic species. Profiling of prokaryotic taxa revealed almost complete assignment to the domain Bacteria, apart from taxa assigned to the domain Archaea, which ranged from 0.2 to 1% in agricultural wastewater substrates.

The eukaryotic community at the phyla level mainly consisted of members of Chlorophyta, Ochrophyta, and Streptophyta (Figure 3). The samples of PigM and CattleM ATS followed by Inoculum II and BGE ATS showed the highest occurrence level of Chlorophyta with abundances of 73%, 60%, 55%, and 40%, respectively. The samples of mWW ATS and Inoculum I displayed comparably lower Chlorophyta content, with up to 22%, but contained the highest levels of Ochrophyta (34 ± 0.4%) and Streptophyta (21 ± 2.7%), respectively. The phylum Chlorophyta consisted mainly of the representatives of Chlorophyceae, Trebouxiophyceae, and Ulvophyceae, with up to 45% (PigM-ATS), 31% (CattleM ATS), and 20% (Inoculum II), respectively (Figure 4a). Streptophyta, mainly from the class Klebsormidiophyceae, were represented with up to 20.6 ± 2.7% and occurred only in the microbial community of Inoculum I. The class representatives of the phyla Ochrophyta comprised Chrysophyceae, Xanthophyceae, Eustigmatophyceae, and Bacillariophyceae, the first two being exclusively present in mWW ATS (with up to 5.9%). The representatives of Eustigmatophyceae were detected as highly abundant in Inoculum II (~12.7%) and in a low proportion (<0.2%) in mWW ATS. The representatives of Bacillariophyceae (diatom) were detected in all tested samples at different abundances, with CattleM ATS showing with about 26% as the highest and PigM ATS with <1% the lowest values (Figure 4a). Noteworthy, only the ATS operated with CattleM, PigM, and BGE wastewater showed increased abundance of the fungal kingdom, such as the phyla Aphelida, Ascomycota, and Mucoromycota (Figure 3).

Analysis of the prokaryotic community of the different wastewater substrates showed that agricultural wastewaters have a very high proportion of Bacillota (formerly Firmicutes, [50]) in the range of 52–72%, followed by the representatives of Bacteroidota (formerly Bacteroidetes) with 12–18% (Figure 3, B for CattleM, PigM, and BGE). Among the Bacillota, Clostridia predominated at the class level (Figure 4b), which is consistent with findings from the literature [55,56]. Highly concentrated municipal wastewater showed the highest taxonomic diversity (Figure 1, Bacteria, mWW) and consisted of high proportions of Pseudomonadota (formerly Proteobacteria), with up to 38%, followed by Bacteroidota (~17%) and Actinomycetota (formerly Actinobacteria, ~11%) (Figure 4, B for mWW). The ATS biofilms cultured with different wastewaters and initial inocula have a similar bacterial composition as the mWW substrate. Pseudomonadota were the most abundant phyla with values ranging from 29 to 56%, with BGE ATS having the lowest and Inoculum II the highest content, followed by representatives of Bacteroidota (5.5% to 19.6%) and Cyanobacteria (4% to 23%) (Figure 3). Among the Pseudomonadota, the proportion of Alphaproteobacteria predominates in BGE ATS (~19%), Cattle ATS (~24%), and Inoculum II (~38%), whereas Gammaproteobacteria prevail with up to 27% in the other microbial communities (Figure 4b).

The presence of cyanobacteria varied among the samples studied, with PigM ATS and CattleM ATS having the lowest levels of around 4%, followed by Inoculum I and BGE ATS with ~10% and mWW ATS and Inoculum II with up to 15.4% and 23.4%, respectively (Figure 3 and Figure 4b).

### 3.4. Microbial Taxa Forming the Wastewater-Driven Biofilm Structures

The compilation of the most abundant taxa at the genus level, consisting of 21 eukaryotic and 32 prokaryotic genera, additionally shows that the studied microbiomes differ significantly from each other as well as from the initial inocula and the applied substrates (Figure 5). Inoculum I is dominated by eukaryotic genera such as *Tritostichococcus* (11.3 ± 0.1%), *Klebsormidium* (20.6 ± 0.3%), and unclassified eukaryotes (38.0 ± 0.3%), whereas Inoculum II comprises *Oedogonium* (13.5 ± 0.1%), *Botryosphaerella* (9.6 ± 0.1%), *Cladophora* (20.1 ± 0.1%), and *Pseudotetraedriella* (12.5 ± 0.02%) (Figure 5a). Therefore, *Klebsormidium*, *Oedogonium*, and *Cladophora* are filamentous or colony-forming microalgae [57], whereas the others are nonfilament-forming. In addition, prokaryotic cyanobacterial genera were highly abundant in the inocula samples, with filamentous taxa such as *Wilmottia* and *Tychonema* found in Inoculum I at 3.4 ± 0.2% and 4.4 ± 0.5%, respectively, whereas the local Inoculum II was dominated by unicellular (non-filamentous) *Altericista* [58] with 23 ± 1.8% of the total prokaryotic features (Figure 5b).

However, no similar distribution was observed in the ATS systems operated with agricultural wastewater, which were largely dominated by unicellular representatives of the Chlorophyta, i.e., Chlorophyceae and Treboxiophyceae. Mostly unicellular and nonfilament-forming genera such as *Edaphochlamys* and *Chlorella*, are highly abundant in PigM ATS and CattleM ATS, with up to 39% and 31%, respectively (Figure 5a). The representatives of Bacillariophyceae, *Cylindrotheca* (~26%), and *Gomphonema* (~15%) are also abundant in CattleM ATS and BGE ATS, respectively. In addition, CattleM ATS and PigM ATS contain comparatively low proportions of cyanobacteria, although BGE ATS were dominated by *Pseudanabaena*, with up to 9.0%. In addition, the ATS systems operated with agricultural wastewater show high levels of *Aphelidium*, an intracellular parasitoid of algae [59], which is particularly abundant in PigM ATS at 17.5 ± 0.1%, followed by CattleM ATS (~5.8%) and BGE ATS (~5.9%). Furthermore, BGE ATS exhibited high amounts of up to 16% of the ascomycetous genus *Ciliophora*, being characterized as an endophytic fungus [60].

In comparison, the mWW ATS community is far more diverse, as shown by alpha diversity (Figure 1), and yet interestingly contains only traces of saprophytic eukaryotes. The photoautotrophic community of mWW ATS consists of non-filamentous genera such as *Tetradesmus* (~10%), *Acutodesmus*, and *Chlorella* (both at ~3.2%, but also filamentous representatives, such as *Dinobryon* (~5.8%) and *Tribonema* (~2.7%), cyanobacterium *Tychonema* (~12%), and the diatoms *Cyclotella* (~15.4%) and *Gomphonema* (~5.4%) as prevalent taxa) (Figure 5a). In addition, mWW ATS contains up to 13.8% unclassified eukaryotes. The prokaryotic communities of mWW ATS and mWW substrate also appeared to be more diverse compared to other communities (Figure 1, Bacteria) and contained genera such as *Nitrospira*, *Aeromonas*, *Shewanella*, *Rhodoferax*, and *Romboutsia*, which were the most abundant, with up to 5.5% (Figure 5b). Similar results were observed for the initial inocula and ATS communities operated with agricultural wastewater, albeit at varying frequencies. While the community of Inoculum I is largely composed of *Ferrunginibacter* (4.8 ± 0.1%), *Rhodanobacter* (8.6 ± 0.1%), and *Luteolibacter* (5.7 ± 0.03%), the community of Inoculum II is, among others, dominated by *Porphyrobacter* (17.3 ± 0.1%) and *Aeromonas* (8.6 ± 0.01%). The most abundant genera of the bacterial communities of CattleM ATS, PigM ATS, and BGE ATS include *Clostridium* (4.8–17.7%), *Gemmatimonas* (1.7–8.8%), *Romboutsia*, and *Rhodobacter* (1–5%) depending on the ATS system, as well as *Hydrogenophaga* with up to 3% (Figure 5). The bacterial community of the agricultural wastewater substrates differed fundamentally, with only a few exceptions, and was largely characterized by representatives of the Bacillota, with the genera *Clostridium*, *Turicibacter*, and *Terrisporobacter* being particularly abundant. In general, the most predominant bacterial genera of the inocula and agricultural wastewater were in most cases not abundant or even absent in the subsequent ATS biofilms (Figure 5b).

## 4. Discussion

Wastewater remediation via microalgae–bacteria consortia is gaining increasing attention [2,24,28]. In the natural environment, microalgae exist as a part of a complex microbial consortium in which different organisms may considerably influence each other to exploit unique biological functions and/or exchange metabolites [10,12]. ATS system-based wastewater treatment provides a platform [8] where phototrophic (autotrophic) and heterotrophic microorganisms coexist in complex and dynamic phycosphere biofilm structures, often referred to as periphyton [10,24]. Within the periphyton, autotrophs, such as microalgae and photosynthetic bacteria, represent the main producers because of their ability to absorb the energy and convert inorganic carbon and nutrients in wastewater to organic matter. Organic carbon (matter) is converted into inorganic matter by decomposing bacteria, e.g., aerobic bacteria that reduce COD while utilizing wastewater nutrients, including those that cannot be utilized by microalgae [61]. In combined wastewater remediation systems consisting of microalgae and bacteria, the photosynthetic activity of the algae provides oxygen for organic matter oxidation by bacteria, whereas algae in turn utilize CO_2_ produced by bacterial respiration for their growth. Yet, there is a wide spectrum of mutual associations between microalgae and bacteria that are predominantly related to nutrient exchange, phycosphere protection, and increased bioavailability of vitamins, metals, and growth-promoting hormones [11,12,17,62]. These beneficial interrelations allow higher efficiencies in the removal of organic matter and nutrients in wastewater treatment than standalone microalgal or bacterial systems [24,28,61]. However, apart from many useful interactions, some heterotrophs/saprophytes/parasites can have negative influences on the periphytic community by being algicidal and opportunistic pathogens [13,63].

In this study, using high-throughput ITS and 16S rRNA gene amplicon sequencing, we profiled the eukaryotic and prokaryotic communities of the periphytic microbiomes of four ATS systems, which were operated with mWW (municipal, tertiary wastewater), diluted cattle and pig manure and supernatant from a biogas plant effluent (CattleM, PigM, and BGE, respectively). The applied wastewater substrates differed greatly in terms of the inorganic nutrient and organic matter content. While the content of nutrients and COD in municipal wastewater (mWW) was very low, the agricultural wastewaters (PigM, CattleM and BGE) contained up to 60 times more N and P and 350 times higher COD (Table 1). Nevertheless, the wastewaters used were very suitable for the growth of the microbial communities, as they were not nutrient-limited and relatively comparable in terms of the N:P ratio (Table 1). In particular, the high ammonium content within the agricultural wastewater should allow for faster microalgal growth [51] and an improved performance of the respective systems. However, the observed biomass productivity was significantly lower in the BGE ATS, CattleM ATS, and PigM ATS systems at 3.5 ± 0.4, 2.1 ± 0.1, and 1.4 ± 0.1 g m^−2^ d^−1^, respectively, compared to 8.3 ± 1.3 g m^−2^ d^−1^ in mWW ATS (Figure 6). This observation could likely be explained by the high amounts of suspended solids, nutrient overload, or the impact of the wastewater microbials that the agricultural wastewaters contained, which may have affected the periphyton communities and ultimately the biomass productivity. Therefore, the microbial communities of the respective wastewater substrates and the initial inocula were also analyzed for comparison. Two different inocula were used for the ATS seeding: an established biofilm of the ATS system at the research center FZ Jülich (Inoculum I) and a periphytic community from the local environment of the WWTP Lichtenau (Inoculum II). Both inocula communities were chosen with the expectation that an established biofilm could provide a faster and more stable foundation for new periphyton development, and the use of indigenous algal–bacterial consortia may be more efficient in nutrient removal and biomass accumulation [61]. The resulting ATS biofilms differed considerably from both initial inocula in terms of occurrence and abundance of taxa at the phyla, class, or genus level (Figure 3, Figure 4 and Figure 5), suggesting an inocula-independent establishment of the microbial communities. In addition, the mWW ATS periphyton was markedly distinct from the ATS systems operated with agricultural wastewater, although these microbiomes showed similarities to one another. As expected, the agricultural wastewater substrates were mainly composed of anaerobes, mostly dominated by the phyla Bacillota [55,56]; however, the prokaryotic communities of the CattleM ATS, PigM ATS, and BGE ATS biofilms resembled the community structure of the mWW ATS and eventually mWW (Figure 2 and Figure 3). The most abundant prokaryotic taxa were similar to the bacteria mostly associated with microalgae in aquatic habitats, which are mainly aerobic heterotrophs from the main groups of Pseudomonadota (mainly α- and γ-proteobacteria) and Bacteroidota [54,64,65]. This observation thus suggests a general adaptation/establishment of the prokaryotic ATS communities to the aquatic habitat, which, however, is still under the influence of the respective substrate due to slight differences between the microbiomes (Figure 3).

Bacteria that reside with microalgae seem to be limited to certain taxa, which is often reflected by the capability of these bacteria to assimilate specific organic carbon sources produced by the microalgae, to cope with antibacterial compounds released by the microalgae, or to fulfil pivotal metabolic roles as nitrifiers or polyphosphate-accumulating organisms (PAOs) [12,13,53,66]. The analyzed periphytic communities contain primary heterotrophs that are metabolizing a diverse spectrum of carbon and energy sources produced by the microalgae, such as polysaccharides, proteins, lipids, fatty acids, cellulose, and aromatic compounds [53]. Interestingly, the bacterial ATS communities also harbor genera with higher metabolic flexibility that, in addition to heterotrophic growth on organic compounds, are also capable, for instance, of aerobic anoxygenic photosynthesis, an ancient form of photosynthesis that does not generate oxygen [67]. Bacteriochlorophyll-synthesizing, anoxygenic photoautotrophs such as *Rhodobacter* and *Porphyrobacter* were enriched in BGE ATS and CattleM ATS (up to 4.9% and 3.4%, respectively) and to a lesser extent in PigM ATS, followed by mWW ATS (<1%) (Figure 5b). In addition, the ATS periphytons are inhabited by potentially strict anaerobic acetogenic microbes, e.g., representatives of the genera *Clostridium* and *Romboutsia* (Figure 5b), that can grow autotrophically utilizing CO_2_ and H_2_ as carbon and energy sources via the reductive acetyl-coenzyme A (Acetyl-CoA) pathway (also known as the Wood–Ljungdahl pathway) [68,69]. Nitrate is the dominant form of biologically available nitrogen in aquatic ecosystems, and bacterial genera such as *Nitrospira* and *Rhodoferax*, commonly involved in nitrification and denitrification processes [53,70], were comparatively enriched in mWW ATS (Figure 5b). In contrast, polyphosphate-accumulating genera such as *Gemmatimonas* [71] were found strongly enriched up to 8.8% in agricultural ATS systems, especially BGE ATS. However, especially in oligotrophic environments, various members of green microalgae, diatoms, and cyanobacteria were found to accumulate intracellular polyphosphate for nutrient scavenging or as an energy reserve for adaptation and survival [66]. This metabolic diversity nicely reflects the fact that the periphytic community of an ATS system displays a highly complex structure consisting of aerobic and anaerobic/anoxic zones, where different microorganisms can find their niches and influence each other [24,28].

Another interesting aspect of the bacterial community analysis was the presence of bacterial genera such as *Aeromonas* and *Shewanella*, which were comparatively abundant in mWW-ATS (Figure 5b) and are known to produce algicidal compounds [72,73]. The inhibitory effect of algicidal compounds secreted by *Aeromonas*, depending on the nutritional status, was observed against the cyanobacteria *Microcystis*, *Anabaena*, and *Pseudanabaena*, as well as the green microalgae *Scenedesmus* and *Chlorella* [74,75]. The genus *Shewanella* is well known for its high metabolic versatility, bioremediation potential, and algicidal activity, specifically against dinoflagellates without affecting chlorophytes, cryptophytes, and diatoms [73,76,77]. Previous research suggests that cell concentration-dependent quorum sensing (QS), a communication mechanism between microorganisms through release, “recognition”, and response to signaling molecules, determines the symbiotic relationships between them [72,73]. However, microalgae can interfere with or inhibit bacterial QS by producing compounds capable of interfering with bacterial signaling receptors and/or response regulators, i.e., lumichrome, a derivative of vitamin B_2_ (riboflavin) secreted by *Chlamydomonas* [78,79]. In this context, the presence of algicidal bacteria within the mWW ATS could be responsible for the relatively low abundance of certain chlorophytes such as *Chlamydomonas*, *Edaphochlamys*, and *Chlorella*, as well as cyanobacteria *Altericista* and *Pseudanabaena* (Figure 5). In fact, the most prominent difference between the profiled biofilms was that mWW ATS harbored many filamentous (benthic) eukaryotic microalgae with up to 10%, whereas the agricultural wastewater-fed ATS systems were with values below 0.1% rather devoid of benthic taxa and were inhabited by chlorophyte genera, such as *Edaphochlamys*, *Chlamydomonas*, and *Chlorella* (40–73%), commonly known to be unicellular and not to form filaments (Figure 5a and Figure 6, microscopic pictures, and bar charts). Interestingly, in a previous study, we isolated and characterized the wastewater-borne, fast-growing microalga *Edaphochlamys* sp. Ck (former *Chlamydomonas* sp.) from pig manure effluent and efficiently used the strain as a mono-substrate for biogas/biomethane generation via anaerobic fermentation [15]. Similarly in this study, *Edaphochamys* was detected at high abundance, with up to 39% in the PigM ATS system fed with diluted pig manure wastewater (Figure 5a), indicating a substrate-specific preference for growth of certain microalgal genera.

The content of diatoms was quite high with 16.8 ± 0.3%, 21.4 ± 0.1%, and 26.0 ± 0.1% in the communities of BGE ATS, mWW ATS, and CattleM ATS systems, respectively, whereas their contribution was rather low, with approximately 0.5%, in PigM ATS (Figure 6). Diatoms are known to play a pivotal role in biofilm architecture due to their ability to adhere to surfaces or aquatic organisms using a mucilaginous *stalk* that is secreted by the cell to form entangled chain-like microcolonies and to facilitate the adherence of other organisms to any substrate [80,81]. Some diatoms were assumed to be the main producers of the mucilage polysaccharide matrix and thus an important contributor to the mucilage-associated phytoplankton community, e.g., genera such as *Cylindrotheca* (main diatom in the CattleM ATS community, Figure 5a), which can release high levels of polysaccharides [82]. Chain formation, mucilage production, aggregation, and/or adherence of *Diatoma* to benthic microorganisms (epiphytic diatoms) may improve nutrient uptake, protect against grazing, and reduce sinking [10,81]. For instance, stalked *Diatoma*, such as Gomphonema, observed in the periphytic communities of mWW ATS and BGE ATS, are known *epiphytes on* Oedogonium, Cladophora, and Tribonema [10,83], which were only detected in the mWW ATS system (Figure 5a). Cyanobacteria could serve as an alternative “platform” for epiphytic diatoms, because of their ability to form benthic mats, in which the individual cyanobacterial cells remain attached after dividing, to form chains of interconnected cells called “trichomes” [84]. Cyanobacterial trichomes were detected in mWW ATS with 14.5 ± 0.6%, followed by BGE ATS with 10.1 ± 0.9%, whereas CattleM ATS and PigM ATS had the lowest content, with up to 4.3%. In addition to the high content of unicellular microorganisms and rather low levels of filamentous microorganisms, the agricultural ATS systems contained high proportions of saprophytic or parasitic organisms (6.6 ± 0.1%, 17.7 ± 0.5%, and 23.6 ± 0.9% for CattleM ATS, PigM ATS, and BGE ATS, respectively (Figure 6, bar charts). These saprophytes/parasites were dominated mostly by two genera, the endophytic fungi *Ciliophora* [60] and an intracellular algal parasitoid *Aphelidium* [59]. In contrast, mWW ATS revealed only minor amounts of saprophytic organisms such as Fungi (<0.4%), Oomycetes (1.9 ± 0.1%), and Metazoa (3.3 ± 0.1%), whereas algal parasitoids such as *Aphelidium* occurred only in traces (<0.03%) (Figure 5a).

To date, it is still unclear which factors determine the structure and diversity of the colonizing periphyton community and whether the bacterial/microalgal associations are species-specific [10,65]. Research efforts have been increasingly focused on filamentous microalgae with large colony sizes and indigestible cell walls, such as *Oedogonium* and *Tribonema*, because of their evident advantages in wastewater treatment over unicellular microalgae [57,85]. The advantages of (filamentous) microalgae–bacteria symbiosis, ease of harvesting, and resistance to predation by grazers can improve biomass production and nutrient removal efficiency [24,28]. Low levels of filament-forming microorganisms and high parasite load on the biofilm, as observed here in the agricultural ATS systems, might prevent the formation and maintenance of a stable (periphyton) biofilm or hinder the adherence of different species under unfavorable conditions. This is also reflected in the biomass productivity observed during the harvest period for BGE ATS, CattleM ATS, and PigM ATS, which was 58%, 75%, and 83% lower than that of mWW ATS, respectively (Figure 6). The high content of organic matter in agricultural wastewater substrates (Table 1, COD) could promote or favor the proliferation of destruents, which could subsequently have a negative impact on filament-forming microalgae, as increased stimulation of EPS secretion, an essential component of biofilm formation, has been observed under conditions of nutrient or carbon deficiency [28,86].

In addition, a wealth of research points to a close link between the diversity of ecosystems and their stability, with variations in species composition providing the mechanical basis for explaining the relationship between species richness and ecosystem function [10]. Indeed, among the samples tested here, the mWW ATS and mWW substrates showed the highest diversity for both eukaryotic and prokaryotic communities, whereas the lowest diversity for eukaryotes was observed for PigM ATS, and Inoculum II was the least diverse among prokaryotes (Figure 1). This again illustrates that in an open system such as the ATS system, the inoculum does not seem to have made a significant contribution to the biofilms formed; instead, the species diversity appears to be habitat- and substrate-dependent. Thus, higher species diversity seems to increase the adaptability of the biofilm, even under rapidly changing environmental conditions, and allows for stable and higher biomass productivity (Figure 1, Figure 3 and Figure 6).

Future studies will reveal which of the most abundant microalgae and bacterial species detected within the ATS periphytons represent the so-called keystone or indicative species [87,88] that play a crucial role in various species interactions and impact the performance and dynamics of the ATS system. Furthermore, multi-omics analysis of the ATS communities, including metatranscriptome, metaproteome, or metabolome approaches, will provide detailed information on microbial activities in the environment and unravel the fundamentals behind the associations. Increased understanding of the microbial interactions within the dynamics of ATS periphytic microbiomes would enable a more efficient wastewater treatment process and biomass production.

## 5. Conclusions

Wastewater remediation using microalgal–bacterial biofilms is receiving increasing attention. Factors affecting the development, structure, and function of periphytic microbiomes are fundamental to the overall performance of the ATS system; however, little is known about the symbiotic interactions and behavior of community members in response to wastewater variations and environmental conditions.

In this study, we present an amplicon-based assessment of the periphytic communities of four ATS systems operated with municipal wastewater (mWW), diluted cattle and pig manure (CattleM and PigM), and biogas plant effluent supernatant (BGE), which were inoculated with equal parts of two established periphytic biofilms from the research center Jülich (FZJ, Inoculum I) and the local environment at the WWTP Lichtenau (Inoculum II). Surprisingly, the resulting ATS biofilms differed considerably from both the initial inocula in terms of occurrence and abundance of microbial taxa, thus showing an inocula-independent establishment of the communities. The comparison of the biofilm grown with low-concentrated nutrient content, as in mWW, with high nutrient and COD content, as in agricultural wastewater (CattleM, PigM, and BGE, Table 1), revealed significant differences in terms of biodiversity and microbiome structuring. The agricultural biofilms showed rather low microbial biodiversity, were mainly colonized by unicellular microalgae, and contained a high relative proportion of saprophytic and parasitic microorganisms. In contrast, the mWW periphyton was highly diverse for both eukaryotic and prokaryotic communities and contained a large proportion of benthic microalgae with low levels of parasites. Thus, this high biodiversity and the presence of filamentous (benthic) microalgae appear to increase the resilience and adaptability of the biofilm to the prevailing environmental conditions, resulting in a stable and higher biomass productivity. Overall, this study expands our understanding of the microbiome development, structure, and function in dependence on the respective wastewater within the ATS-based wastewater treatment process.

## Figures and Tables

**Figure 1 microorganisms-11-02994-f001:**
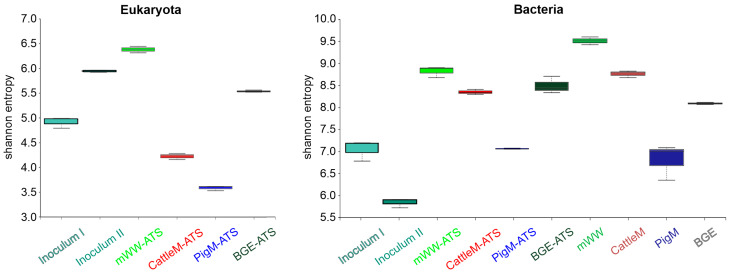
Biodiversity of eukaryotic and prokaryotic communities of initial inocula, different ATS biofilms, and wastewater substrates. Results are presented as the boxes’ bounds and lines representing maxima, medians, and minima; n = 3.

**Figure 2 microorganisms-11-02994-f002:**
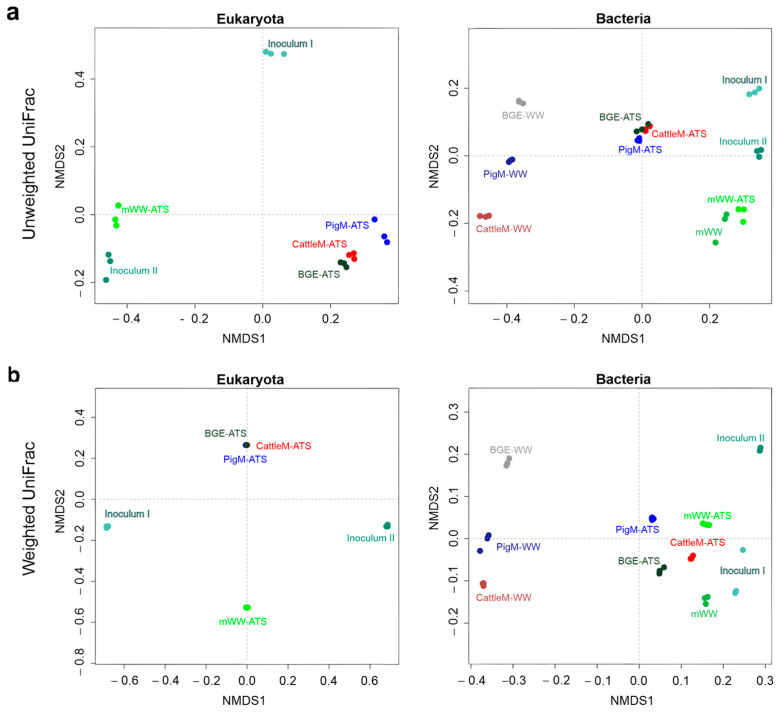
Nonmetric multidimensional scaling (nMDS) of Bray–Curtis distances in eukaryotic and prokaryotic communities of different ATS biofilms, wastewater substrates, and initial inocula. Principal component analyses are based on the (**a**) presence/absence of and (**b**) normalized relative abundance of ASVs and presented as unweighted and weighted UniFrac distance plots, respectively.

**Figure 3 microorganisms-11-02994-f003:**
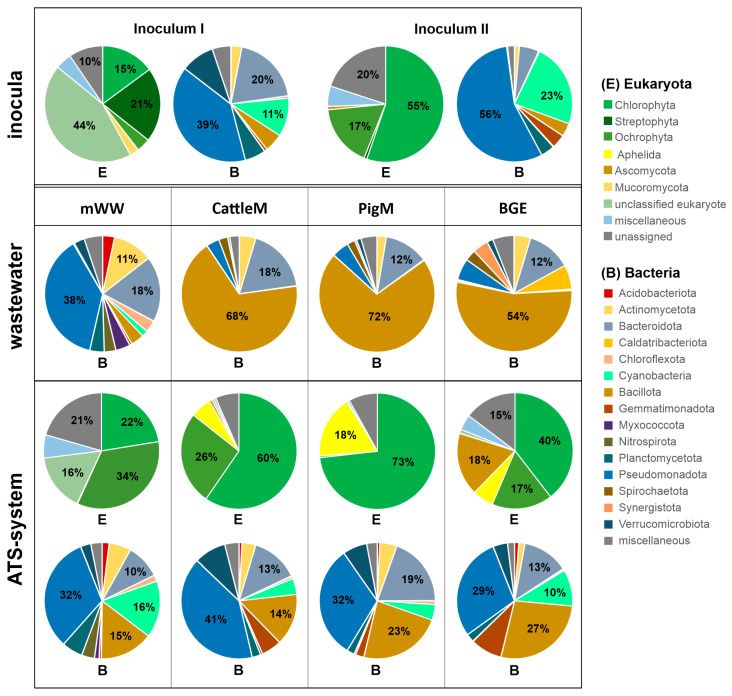
Taxonomic profiles on the phyla level of the initial inocula, the different ATS systems, and their respective wastewater substrates as deduced from 16S rRNA and ITS gene amplicon data. Phyla with a maximal relative abundance of less than 3% were summarized as “miscellaneous”. Taxa that were not taxonomically assigned at the phyla level were summarized as “unclassified”.

**Figure 4 microorganisms-11-02994-f004:**
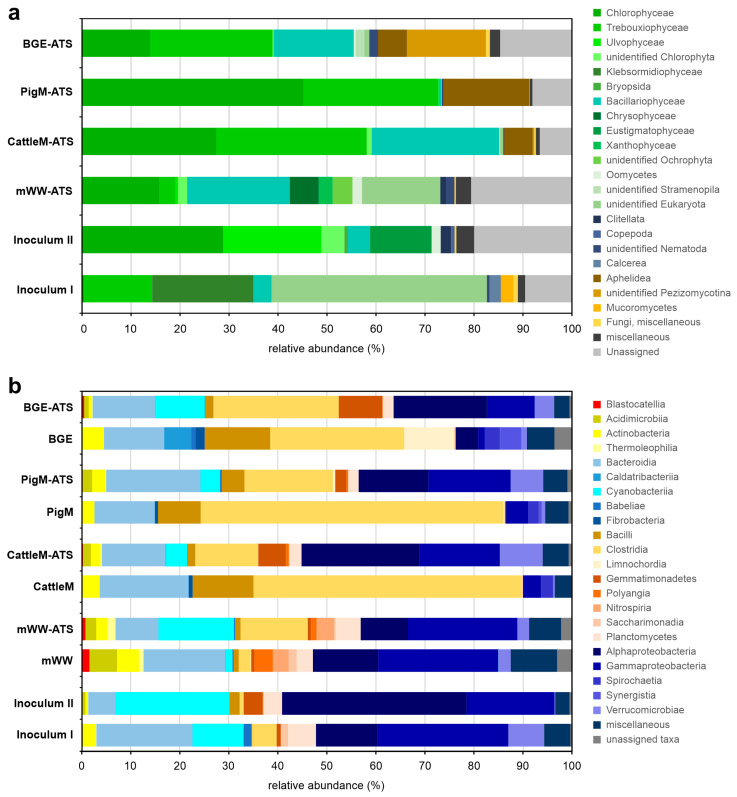
Taxonomic profiles on the class level of the initial inocula and different ATS systems and their respective wastewater substrates as deduced from (**a**) ITS and (**b**) 16S rRNA gene amplicon data. Classes with a maximal relative abundance of less than 1.5% were summarized as “miscellaneous”. Class-level taxa that could not be taxonomically assigned were classified as “unassigned“.

**Figure 5 microorganisms-11-02994-f005:**
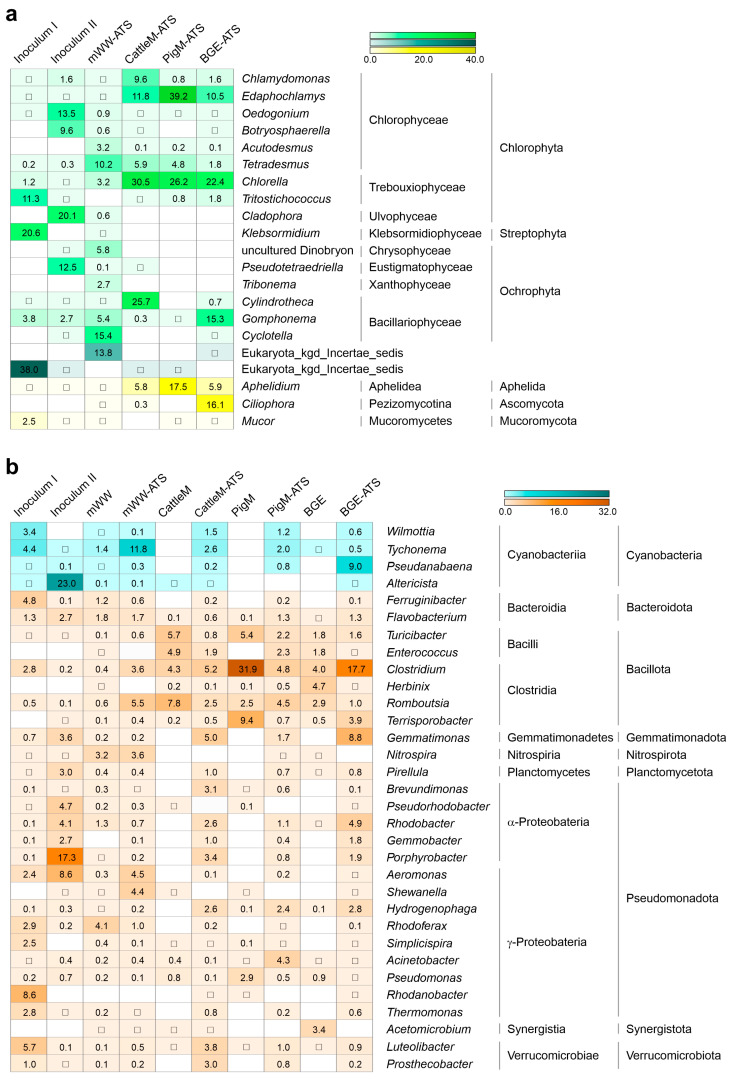
Heatmap of most abundant taxa on genus level of the initial inocula, the different ATS systems, and their respective wastewater substrates. Genera with a maximal relative abundance higher than 2.5% in at least one condition are shown as deduced from (**a**) ITS and (**b**) 16S rRNA amplicons. Detailed information about the statistical analyses is provided in Appendix A. Squares indicate very low abundance appearance (<0.01) of the respective taxa.

**Figure 6 microorganisms-11-02994-f006:**
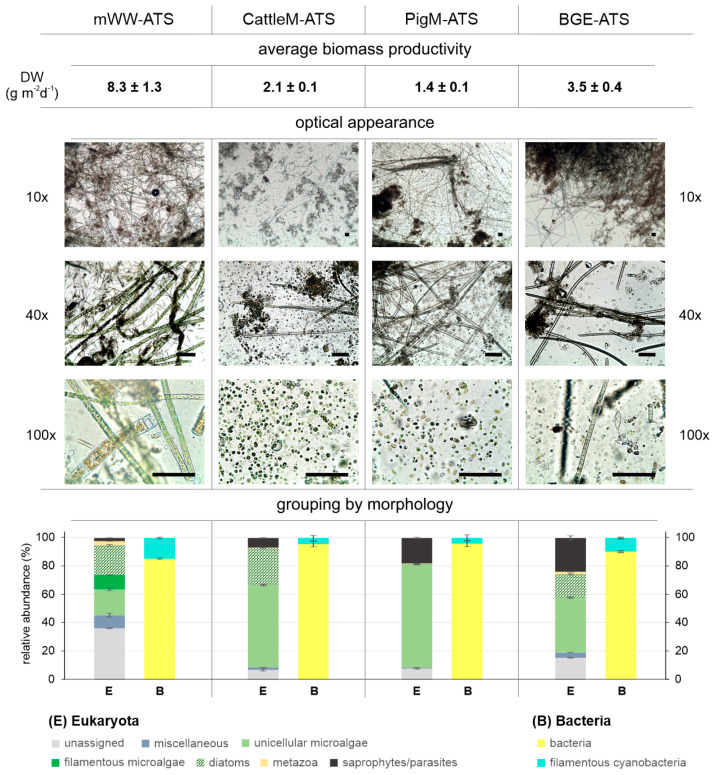
Morphological appearance and grouping of filamentous and unicellular microalgae within the ATS periphytons. Shown are the average biomass productivities of the biofilms from the ATS systems (during two weeks before harvest) operated with municipal wastewater and diluted cattle and pig manure, as well as supernatant from biogas effluent (mWW ATS, CattleM ATS, PigM ATS, and BGE ATS), respectively. Optical microscope images (the scale bar refers to 50 μm length) of the periphytic communities were taken during the sampling for sequencing analysis. The morphological grouping of microbiome members was estimated based on the relative abundance of detected taxa via ITS and 16S rRNA amplicon sequencing results, optical appearance and in agreement with observations from the literature. The grouping results for (E) Eukaryota and (B) Bacteria are presented as bar graphs; SD, n = 3. Error bars for biomass productivity represent SE, n = 7.

**Table 1 microorganisms-11-02994-t001:** Physiochemical properties of the wastewater substrates.

Parameter			mWW	PigM	CattleM	BGE
pH			6.7	7.6	8.0	7.2
COD	mg O_2_	mg L^−1^	<15	1847.6	2638.6	5265.3
Ammonium	NH_4_-N	mg L^−1^	0.8	173.0	50.0	158.5
Nitrate	NO_3_-N	mg L^−1^	1.6	25.4	8.8	11.6
Total Nitrogen	N	mg L^−1^	3.9	207.5	111.0	250.5
Total Sulphur	S	mg L^−1^		15.5	9.5	28.0
Total Phosphorous	P	mg L^−1^	1.6	98.0	49.0	92.5
Orthophosphate	PO_4_-P	mg L^−1^	1.4	23.2	12.7	28.3
Potassium	K	mg L^−1^	16.0	164.5	145.0	353.0
Calcium	Ca	mg L^−1^	40.0	41.0	82.5	142.0
Magnesium	Mg	mg L^−1^	4.0	62.0	27.5	42.0
TS		% *w*/*w*	<0.1	0.21	0.28	0.47
VS		% *w*/*w*	<0.1	0.15	0.21	0.34
N:P molar ratio			5.4	4.7	5.0	6.0

Abbreviations: COD, chemical oxygen demand; TS, total solids; VS, volatile solids; *w*, weight.

## Data Availability

The raw 16S rRNA und ITS amplicon sequences are deposited in the NCBI Sequence Read Achieve (SRA) database with the access number PRJNA1045004.

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
