# Peer review of "Microbial Diversity and Community Structure of Wastewater-Driven Microalgal Biofilms"

_microorganisms, 2023, doi:10.3390/microorganisms11122994_

Round 1

Reviewer 1 Report

Comments and Suggestions for Authors

The manuscript entitled Microbial diversity and community structure of wastewater-driven microalgal biofilmsis aims to periphytic microbiomes development and its symbiotic interactions in algae-bacteria consortia during wastewater remediation in Solar-driven ATS (algal turf scrubber) system.

The manuscript entitled Microbial diversity and community structure of wastewater-driven microalgal biofilms” is aims to periphytic microbiomes development and its symbiotic interactions in algae-bacteria consortia during wastewater remediation in Solar-driven ATS (algal turf scrubber) system. To my mind this manuscript is topical and corresponding to the aims and scopes of the “Microorganisms journal. Manuscript made a very good impression on me. The work was carried out at a high technical level, the research objects were chosen very well. This manuscript can be published without significant changes, it is difficult to add anything to it. I wish the authors good luck and other excellent works in the future.

To my mind this manuscript is topical and corresponding to the aims and scopes of the “Microorganisms journal. Manuscript made a very good impression on me. The work was carried out at a high technical level, the research objects were chosen very well. This manuscript can be published without significant changes, it is difficult to add anything to it. I wish the authors good luck and other excellent works in the future.

Author Response

Dear reviewer, we are very pleased that you acknowledge our work and like to thank you for the positive feedback.

Reviewer 2 Report

Comments and Suggestions for Authors

I think that the article entitled “Microbial diversity and community structure of wastewater driven microalgal biofilms” presents results of interest to both the scientific community and the general population. However, some adjustments are necessary to improve the document.

Abstract

The abstract needs to be rewritten. I think this section should have more data to make it more attractive.

Keywords

I don't think you need so many keywords, please select the main ones.

I think the Materials and Methods section is adequate.

Results and discussion

Physiochemical properties of the wastewater

I think this section is missing an in-depth discussion, it is not just a description of the results L(169-188).

Biodiversity of the different wastewater-driven microbiomes

I think that a comparison should have been made with other works and I need to go deeper, hold discussions, etc.

The quality of all figures is poor, please improve.

The conclusion must be adjusted to the focus of the work

References are adequate

Author Response

Reviewer 2:

I think that the article entitled “Microbial diversity and community structure of wastewater driven microalgal biofilms” presents results of interest to both the scientific community and the general population. However, some adjustments are necessary to improve the document.

Dear reviewer, thank you a lot for your valuable suggestions and we are very pleased that you acknowledge our work. We improved the manuscript based on your suggestions wherever possible.

Abstract: The abstract needs to be rewritten. I think this section should have more data to make it more attractive.

Response to reviewer 2’ comment 1: We have modified some parts of the abstract, as suggested by the reviewer. Unfortunately, we are limited in the number of words by the requirements of the journal, so that a more detailed summary of the important findings is rather difficult to realise.

Keywords: I don't think you need so many keywords, please select the main ones.

Response to reviewer 2’ comment 2: As suggested by the reviewer, we have reduced the number of keywords.

I think the Materials and Methods section is adequate.

Response to reviewer 2’ comment 3: No comment, thank you for your valuable review of our manuscript.

Results and discussion

Physiochemical properties of the wastewater: I think this section is missing an in-depth discussion, it is not just a description of the results L(169-188).

Response to reviewer 2’ comment 4: As suggested by the reviewer, we have included more detailed data analysis into the result chapter (Lines 169-202) and complemented the discussion section (Lines 413-424).

Biodiversity of the different wastewater-driven microbiomes: I think that a comparison should have been made with other works and I need to go deeper, hold discussions, etc.

Response to reviewer 2’ comment 5: As suggested by the reviewer, we have extended the respective subchapter (Line 226-229). Furthermore, we had already compared the bacterial communities of the ATS biofilms and the respective wastewater substrates with aerobic and anaerobic microbiomes from other publications in our discussion chapter (lines 437-444).  In addition, we compared the microalgal communities (filamentous and non-filamentous as well as periphytic) in our system with those already published in several places (lines e.g. 518-536, 547-554). A further, more detailed comparison based on published data from other publications is beyond the scope of this paper.

The quality of all figures is poor, please improve.

Response to reviewer 2’ comment 6: The authors are thankful for bringing this issue to our attention. We have included all figures in high resolution format (in TIFF) into the supplementary zip-file.

The conclusion must be adjusted to the focus of the work

Response to reviewer 2’ comment 7:  We have modified the conclusions, as suggested by the reviewer.

References are adequate

Response to reviewer 2’ comment 8: No comment, thank you for your valuable review of our manuscript.

Round 2

Reviewer 2 Report

Comments and Suggestions for Authors

I think the manuscript can be published in its current version.

Thank you very much.